# Adoption of the voluntary conflict of interest statement on PubMed

Stephanie Rogus[1], Joseph S. Ross[2,3,4], Peter Lurie[1]*

1 Center for Science in the Public Interest, Washington, DC, United States of America, 2 Center for Outcomes Research and Evaluation, Yale New Haven Hospital, New Haven, Connecticut, United States of America, 3 Section of General Internal Medicine and the National Clinical Scholars Program, Department of Internal Medicine, Yale School of Medicine, New Haven, Connecticut, United States of America, 4 Department of Health Policy and Management, Yale School of Public Health, New Haven, Connecticut, United States of America

* plurie@cspinet.org

**Data Availability Statement:** All data files are available from the Dryad database (reviewer URL: https://datadryad.org/stash/share/zZLdgW-pVjwxzhH6Y07sk-eUee_Egzrq_Q3dwK0pGDc; DOI: doi:10.5061/dryad.h70rxwds8).

## Abstract

In 2017, the National Library of Medicine (NLM) added a voluntary field for conflict of interest (COI) statements ("posted COI") on the abstract page of PubMed, but the extent to which it is used is unknown. This repeated cross-sectional study examined journals and articles indexed on PubMed from 2016 through 2021. We described the proportion of all journals with at least one article that included a posted COI and the percentage of all articles that included a posted COI over time. We also examined 100 randomly selected articles published between June 2021 and May 2022 from each of the 40 highest impact journals. For these, we established whether the articles had published COIs, and, of these, the proportion that included a posted COI. Among approximately 7,000 journals publishing articles each year, the proportion of journals with at least one article with a posted COI statement increased from 25.9% in 2016 to 33.2% in 2021. Among nearly 400,000 articles published each year, the proportion of articles that included a posted COI also increased from 9.0% in 2016 to 43.0% in 2021. Among 3,888 articles published in the 40 highest impact journals in 2021–2022, 30.2% (95% CI: 28.7%-31.6%) had published COIs; of these, 63.3% (95% CI: 60.4%-66.0%) included a posted COI. Use of the PubMed COI statement has increased since it became available in 2017, but adoption is still limited, even among high impact journals. NLM should carry out additional outreach to journals that are not using the statement to promote greater transparency of COIs.

## Introduction

Every year, industry spends an estimated $60 billion funding research on drugs, biotechnology, and medical devices [1]; commercial company expenditures on phase 1–3 clinical drug trials outstrip the National Institutes of Health's spending on trials [2]. Disclosure of such funding is important in part because a Cochrane systematic review on industry sponsorship of research found that industry-funded studies tend to produce results favorable to their company sponsors more often than studies not funded by industry [3]. It concluded that this

**Funding:** This research was supported by the Harvey Motulsky and Lisa Norton-Motulsky fund. The funders had no role in study design, data collection and analysis, decision to publish, or preparation of the manuscript.

**Competing interests:** SR and PL have declared that no competing interests exist. JR has read the journal's policy and has the following competing interests: currently receives research support through Yale University from Johnson and Johnson to develop methods of clinical trial data sharing, from the Food and Drug Administration for the Yale-Mayo Clinic Center for Excellence in Regulatory Science and Innovation (CERSI) program (U01FD005938), from the Agency for Healthcare Research and Quality (R01HS022882), and from Arnold Ventures; formerly received research support from the Medical Device Innovation Consortium as part of the National Evaluation System for Health Technology (NEST) and from the National Heart, Lung and Blood Institute of the National Institutes of Health (NIH) (R01HS025164, R01HL144644); was an expert witness at the request of Relator's attorneys, the Greene Law Firm, in a qui tam suit alleging violations of the False Claims Act and Anti-Kickback Statute against Biogen Inc. that was settled September 2022; and is currently a Deputy Editor at JAMA, was formerly the U.S. Outreach and Research Editor at the BMJ from 2020-2023, and was formerly an Associate Editor at JAMA Internal Medicine from 2013-2019. This does not alter our adherence to PLOS One policies on sharing data and materials. There are no patents, products in development, or marketed products associated with this research to declare.

finding may be partly caused by the choice and dosing of comparators, selection of outcomes, and selective analysis, reporting, and publication. It is also possible that industry is successfully identifying research projects that are more likely to yield results favorable to their products.

The National Academy of Medicine defines a conflict of interest (COI) as "circumstances that create a risk that professional judgements or actions regarding a primary interest will be unduly influenced by a secondary interest [4]." In the case of research COIs, the primary interest is typically research integrity, while the secondary interest is generally financial gain. The conflict may not necessarily compromise research integrity, but conflicts increase that risk or create an appearance of risk [5]. Disclosing COIs in biomedical research may allow scientists, medical professionals, and the public to better assess the credibility of scientific findings and explicitly consider the potential impact of funding on the research.

PubMed is a free resource that provides access to a database of citations to published research, along with their abstracts, primarily in the biomedical and health fields, though it includes related disciplines like behavioral and life sciences and bioengineering. It was developed and is maintained by the National Center for Biotechnology Information at the National Library of Medicine (NLM), an institute within the National Institutes of Health [6].

In March 2016, the Center for Science in the Public Interest, along with five other organizations, 62 scientists and physicians, and 5 U.S. senators, asked the NLM to list researchers' COIs in a standardized statement as part of the abstracts indexed on PubMed [7, 8]. Including COIs in the abstract page on PubMed is important because readers may not have access to the full article and many readers rely on the abstract alone to decide whether to read the full text of the article, to draw conclusions about a study, and/or to guide clinical care decisions [9, 10]. A year later, NLM announced it would begin including voluntary COI statements on PubMed, when these statements are provided by the journal, for inclusion in a "Conflict of Interest statement," located just below the abstract on PubMed [11]. These statements may also include declarations of "no conflict." The changes were implemented on March 8, 2017.

Although most journals have COI disclosure policies for authors [12, 13], the onus is on journals to ensure that information from each article is included on PubMed. When journals add their articles to PubMed electronically, they are required to submit and tag 13 pieces of information (e.g., article title, journal, and publisher), whereas 37 other tags are optional or required only if applicable (e.g., COI statement and abstract) [14]. The prevalence of use of the COI statement on PubMed remains unknown. To address this research gap, this study aims to describe the prevalence of the voluntary COI statement in articles indexed on PubMed.

## Methods

### Study design

We conducted a repeated cross-sectional study to assess: 1) the proportion of journals with at least one article that included a COI statement on PubMed and the proportion of articles that included a COI statement on PubMed from 2016–2021; 2) among high impact journals that disclosed a COI anywhere in the published version of the article, the proportion that posted a COI statement on PubMed and article characteristics associated with such posting. IRB review was not required for this study because it is not considered human subjects research according to the U.S. Department of Health and Human Services guidelines.

### Study sample

With assistance from a staff scientist at NLM, we first identified journals indexed on PubMed from January 1, 2016, through December 31, 2021, that published original research (e.g., clinical trials, observational studies, and validation studies), reviews (e.g., systematic reviews, meta-

analyses, and narrative reviews), and comments (e.g., letters, editorials, consensus statements, case reports, and guidelines). We excluded books and documents, legislation and government publications, article preprints, news articles, and datasets on the ground that journal COI disclosure standards for these are less well developed. For each year, we recorded the total number of journals indexed and the number of journals with at least one article that included a COI statement on PubMed (hereafter, "posted COI statement"), indicating that the journal at least sometimes used the COI field. A posted COI statement included any stated COIs or any affirmative statement that there were no COIs in the paper. We also recorded the total number of published articles and the number of those that included a posted COI statement by year. The year 2016 was included to examine the change in use of the COI statement after it was added to PubMed in 2017. These data were collected in January 2023.

We then identified the 40 highest impact journals in 2021, based on Journal Citation Reports [15], that were indexed on PubMed and had published at least 50 articles meeting the study inclusion criteria from June 1, 2021, through May 31, 2022. We collected characteristics of these journals by searching journal, International Committee of Medical Journal Editors (ICMJE) [16], and Committee on Publication Ethics (COPE) websites [17]. Journal characteristics included whether the journal was associated with a professional society or not (society-based/other), journal type (predominantly basic science/predominantly clinical/review/other), journal publication country (U.S./U.K./other), impact factor (continuous), ICMJE and COPE membership (yes/no), and publisher (professional society/Elsevier/Springer/other).

We used a random number generator [18] to randomly select 100 articles from each of the 40 highest impact journals. We examined articles from the top 10 journals (993 articles, as one journal had only published 93 articles) to inform our final sample size, revealing that 33% included a COI in the article itself and 54% of those included a posted COI statement. With about 4,000 articles (40 journals) overall, we would be able to detect that prevalence of posted COI statement use among those with a published COI with a margin of error of +/- 3% at a 95% confidence level [19].

To determine what fraction of all COIs were reflected in posted COI statements, we downloaded and manually searched articles for COIs (a financial link with the topic under investigation) anywhere in the main text of the article, including in the funding, disclosures, and acknowledgements fields (hereafter, "published COIs") and documented the location of all disclosures. We downloaded these articles between June 2022 and August 2022. Published COIs located in a disclosures field or COI statement in the article were coded as "COI statement" and those located in the acknowledgements or funding fields were coded separately. We included relevant published COIs regardless of when they took place. If the article declared "no conflict of interest" and there were no published COIs found in the main text of the article, we did not consider the article to have published COIs. We also recorded the study type of each article (comment/original research/review). We then assessed the PubMed entry corresponding to each article with published COIs to determine if the published COIs were described in a posted statement. A flow chart that shows how the samples were selected is included in Fig 1.

Two members of the research team reviewed all articles in the first seven of the 40 journals to establish consistency in coding (about 700 articles). The reviewers then compared notes to identify inconsistencies in their identification of COIs and reached consensus through discussion. With the exception of about 10 articles, there was agreement between the two reviewers on identifying articles with published COIs and posted COIs, COI location, and study type. Thereafter, to conserve resources, one reviewer examined the remaining journals.

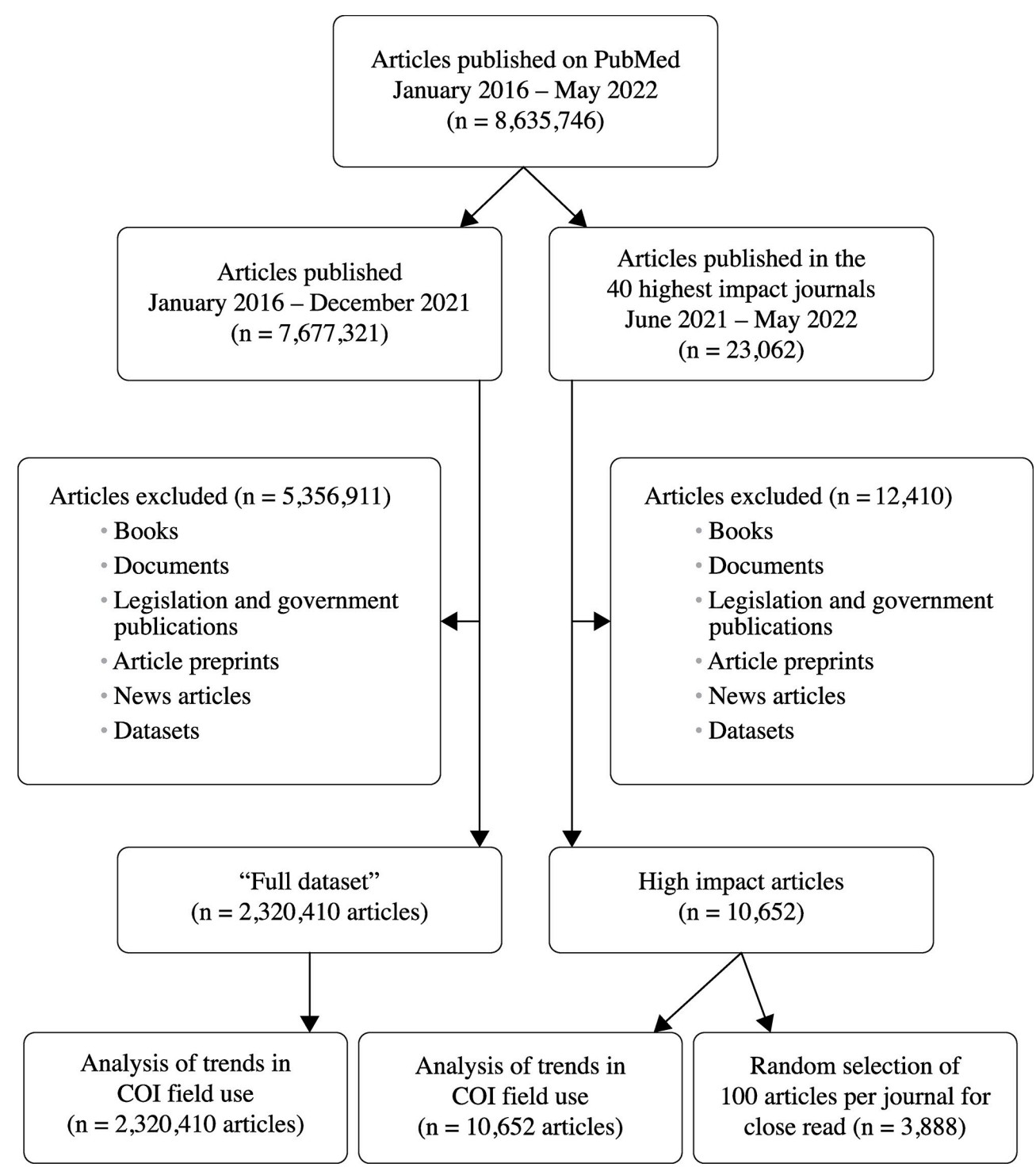

**Fig 1. Article selection flow chart.**

## Statistical analysis

We reported descriptive statistics showing the prevalence of all journals with at least one article that included a posted COI statement and of all articles that included a posted COI statement

each year. Failure to publish at least one article that used the posted COI statement in a given year is indicative of non-use of the statement by that journal.

We described the number and proportion of articles from the top 40 high impact journal subset with published COIs that also included a posted COI statement and calculated their exact 95% confidence intervals (CIs), providing one-sided CIs when the prevalence was 0% or 100%. We also examined the correlation between this proportion and journal impact factor with a Spearman's Rank Correlation Test. Finally, we constructed bivariate odds ratios and a multivariate logistic regression to examine whether published COI location and study type (independent variables) were associated with the inclusion of a posted COI statement (dependent variable).

We conducted the analysis using STATA version 17 [20], and differences were considered statistically significant if the P-value (2-sided) was less than 0.05.

## Results

### 1. Analysis of the full dataset (2016–2021)

Although the COI policy only went into effect in 2017, some posted COI statements were added retroactively (Jeff Beck (NLM), oral communication, April 8, 2022). The proportion of journals on PubMed with at least one article with a posted COI statement increased from 25.9% in 2016 (1,790/6,894 journals) to 33.2% in 2021 (2,511/7,550 journals), although that increase has not been monotonic (Fig 2). At the article level, the percentage with a posted COI statement steadily increased, from 9.0% in 2016 (31,718/350,587 articles) to 43.0% in 2021 (186,500/433,386 articles).

### 2. Analysis of the 40 highest impact journals

**Journal characteristics (June 2021 –May 2022).** Most of the 40 highest impact journals were not run by a society (72.5%) and most published predominantly clinical research (42.5%) and reviews (35.0%) (Table 1). All but one was based in the U.S. (32.5%) or the U.K. (65.0%) and the median 2021 impact factor was 65.9 (range: 43.4–202.7). Most journals were not ICMJE members (60.0%) but 95.0% of journals were members of COPE. Seventy-five percent of journals were published by Elsevier (32.5%) or Springer (42.5%).

Between June 1, 2021, and May 31, 2022, all of the 40 highest impact journals published at least one article with a posted COI statement, indicating widespread awareness and ability to use the COI field.

**Prevalence of posted COI statements (2016–2021).** Of the 40 highest impact journals in 2021, the number that published at least one article with a posted COI statement ranged from 84.6% (33/39 journals) in 2016 (one journal did not publish any articles that year) to all 40 in 2021 (Fig 3). At the article level, the percentage with a posted COI statement increased over time from 5.9% in 2016 (706/11,936 articles) to 35.1% in 2021 (3,725/10,598 articles).

**COI posting prevalence (June 2021 –May 2022).** Table 2 lists the 40 journals with the highest impact factors in 2021, along with the percentages of the randomly selected 100 articles per journal that contained published COIs and, among those, the percentage that included posted COIs. Six of the 40 journals published fewer than 100 articles between June 1, 2021, and May 31, 2022, for a total of 3,888 articles. The percentage of articles with published COIs by journal ranged from 5.0% to 83.0% [median: 26.0%; interquartile range: 12%-37.5%] and the percentage of articles with published COIs that also included a posted COI statement ranged from 0.0% (Science [7 articles with published COIs] and Nature Materials [5 articles with published COIs]) to 100.0% (BMJ, Lancet Psychiatry, Lancet Public Health, Cell, Cell Research, and Immunity) [median: 48.1%; interquartile range: 25.9%-94.0%]. A Spearman's Rank

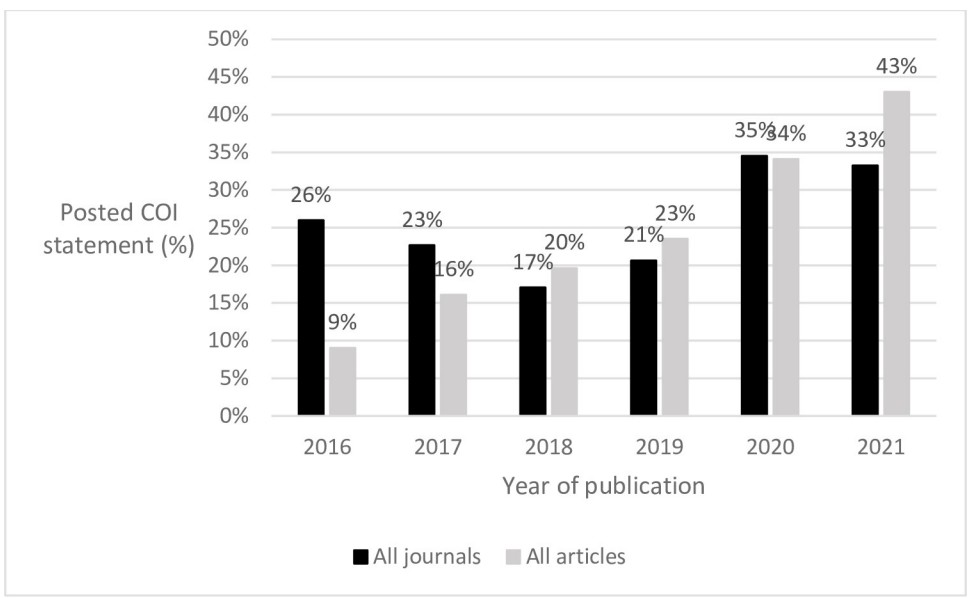

**Fig 2. Percentage of all journals that included a posted COI statement for at least one article and the percentage of all articles with a posted COI statement, 2016–2021, as of 2023.**

Correlation Test indicated that there was no correlation between journal impact factor and the percentage of articles with published COIs that included a posted COI statement within this subset (r = -0.0854; p = 0.6001).

Of all 3,888 randomly selected articles across the 40 highest impact journals, 69.8% [95% CI: 68.3%-71.2%] did not have published COIs. Of the 1,174 articles with published COIs, 63.3% [95% CI: 60.4%-66.0%] included a posted COI statement (Fig 4). (An analysis of posting and publication rates weighted by impact factor provided essentially identical results and is not presented).

**Characteristics and predictors of COI posting (June 2021 –May 2022).** Of the subset of articles with published COIs (n = 1,174), 27.0% were original research, 33.0% were review articles, and 39.9% were comments (Table 3). Most (n = 1,036, 88.2%) of these articles had published COIs in only the COI statement in the main text of the article, 7 (0.6%) had published COIs in only the acknowledgements, and 131 (11.2%) had published COIs in more than one location. None of the conflicts that appeared in an acknowledgements or funding section in the paper appeared in a posted COI statement.

The bivariate regression results show that, among articles with published COIs, the odds of original research articles including a posted COI statement were over 3 times those odds for comment articles (OR: 3.74; 95% CI: 2.63, 5.32), an association that rose to over 5-fold (OR: 5.40; 95% CI: 3.51, 8.32) in the multivariate analysis. In the multivariate analysis, articles with published COIs in more than one location in the article text were 60% less likely than articles with published COIs in one location only to have a posted COI (OR: 0.40; 95% CI: 0.24, 0.68) (Table 3).

## Discussion

This examination of the adoption of the COI statement on PubMed since its inception in 2017 found that, while the use of the posted COI statement on PubMed has increased fairly steadily, overall use of the statement could be improved. In 2021, only 33% of journals used the

**Table 1. Characteristics of the 40 highest impact journals.**

| Journal Characteristic | Number (%) of Journals |
|---|---|
| Society | |
| Society-based | 11 (27.5) |
| Other | 29 (72.5) |
| Journal type | |
| Basic science | 6 (15.0) |
| Clinical research | 17 (42.5) |
| Review | 14 (35.0) |
| Other | 3 (7.5) |
| Country/Region | |
| U.S. | 13 (32.5) |
| U.K. | 26 (65.0) |
| Other | 1 (2.5) |
| Median 2021 impact factor | 65.9 |
| ICMJE membership | |
| Yes | 16 (40.0) |
| No | 24 (60.0) |
| COPE membership | |
| Yes | 38 (95.0) |
| No | 2 (5.0) |
| Publisher | |
| Society | 8 (20.0) |
| Elsevier | 13 (32.5) |
| Springer | 17 (42.5) |
| Other | 2 (5.0) |

statement for at least one article. Among the 40 highest impact journals, use of the COI statement on PubMed was higher, with all journals posting COI statements for at least one article in 2021 but only 58% of articles with known COIs using the posted COI statement. Based on whether journals used the posted COI statement at least once per year, higher impact journals seem to be utilizing the posted statement more often than lower impact journals, but all journals could improve their consistent use of the posted COI statement.

The underutilization of the posted COI statement is a combination of journals that do not use the statement, journals that are using the statement inconsistently, and journals that are using the posted COI statement but failing to transfer conflicts from the acknowledgements or funding sections of their papers.

The first, and largest, problem is that most journals, particularly those with lower impact factors, do not use the posted COI statement at all. Journals have to transfer and tag over a dozen pieces of information for each article, yet many choose not to transfer COI information. Given that journals had five years to adopt the COI statement on PubMed at the time of our analysis, the relatively low adoption rate raises questions about whether all journals are aware of the option. According to the NLM, requiring the use of the posted COI statement as a mandatory condition of listing on PubMed is not advisable because it could limit submissions of journal articles, which does not align with PubMed's mission as a comprehensive resource for finding and indexing publications (Jeff Beck (NLM), email communication, June 15, 2023).

NLM can, however, strongly encourage use of the statement. When the statement was first made available, the agency notified publishers through email and included information about

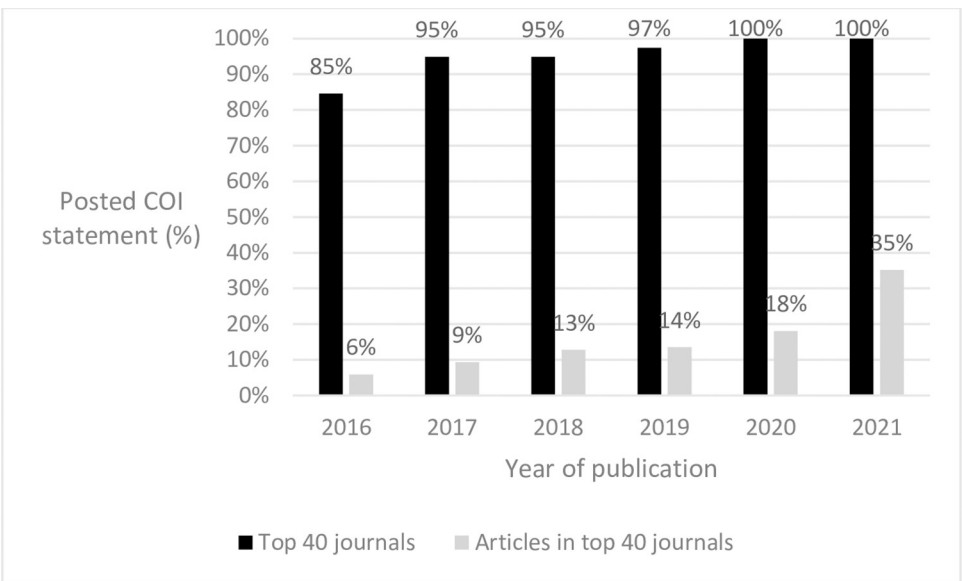

**Fig 3. Among the 40 highest impact journals in 2021, percentage of all journals that included a posted COI statement for at least one article and the percentage of all articles that included a posted COI statement, 2016–2021, as of 2022.**

the statement in a technical bulletin (Jeff Beck (NLM), oral communication, April 8, 2022) [13]. The group JATSforReuse, a voluntary collaboration of publishers and NLM [21], published technical recommendations for pulling COI statements from article metadata in 2020 [22], and, according to NLM, the agency continues to encourage publishers to include COI statements on PubMed (Jeff Beck (NLM), email communication, June 15, 2023). But NLM should go further and individually notify journals that did not use the posted COI statement in the most recent year.

We examined the second problem, that of inconsistent posted COI statement use, through our collection of a random subset of articles from the 40 highest impact journals. Journals varied in their use of the posted COI statement. In our sample, Science and Nature Materials never used the posted statement, although they have used the statement in articles not included in our sample, while only five journals used posted COI statements every time there was a published COI.

The third problem, which is much less common than the first two and the most difficult to remedy, relates to the transfer of conflicts located in an acknowledgements or funding section of an article to the posted COI statement. Of the articles in the top 40 journals, none with published COIs in an acknowledgements or funding section in the paper included those COIs in the posted COI statement, suggesting that journals pull data only from the COI statement in the article. In some instances, study funding was included within the abstract itself on the abstract page of PubMed. But this study was focused on the use of the COI statement, so these disclosures were not included in our analysis. Currently, including disclosures from acknowledgements and funding sections of published articles would require some degree of manual curation and may therefore be difficult to accomplish in practice. Standardization of disclosure practices to use a single field across all journals would greatly facilitate data transfer.

This study had several limitations. First, for the full dataset, we established the prevalence of posted COI statement use in part by determining whether the journal used the COI field at least once in a given year. Measuring the actual number of postings, as we also did, is

**Table 2. Articles with published COIs and a posted COI statement among high impact journals, June 2021 through May 2022.**

| Journal name | Impact factor | # Articles randomly selected to be hand-searched, 3,888 | Articles with published COIs, %; 95% CI | Articles with published COIs that included a posted COI, %; 95% CI |
|---|---|---|---|---|
| Lancet | 202.7 | 100 | 31.0; 22.1, 41.0 | 74.1; 55.3, 88.1 |
| New England Journal of Medicine | 176.1 | 100 | 37.0; 27.5, 47.2 | 2.7; 0.0, 14.1 |
| Journal of the American Medical Association (JAMA) | 157.3 | 100 | 38.0; 28.4, 48.2 | 34.2; 19.6, 51.3 |
| Nature Reviews Molecular Cell Biology | 113.9 | 100 | 11.0; 5.6, 18.8 | 18.1; 2.2, 51.7 |
| Nature Reviews Drug Discovery | 112.3 | 93 | 36.5; 26.8, 47.1 | 52.9; 35.1, 70.2 |
| Nature Reviews Immunology | 108.5 | 100 | 24.0; 16.0, 33.5 | 70.8; 48.9, 87.3 |
| Lancet Respiratory Medicine | 102.6 | 100 | 64.0; 53.7, 73.3 | 90.6; 80.7, 96.4 |
| British Medical Journal | 93.3 | 100 | 11.0; 5.6, 18.8 | 100.0; 71.5, 100.0 |
| Nature Medicine | 87.2 | 100 | 55.0; 44.7, 64.9 | 25.4; 14.6, 39.0 |
| Lancet Microbe | 86.2 | 100 | 24.0; 16.0, 33.5 | 91.6; 73.0, 98.9 |
| Nature Reviews Microbiology | 78.3 | 100 | 12.0; 6.3, 20.0 | 33.3; 9.9, 65.1 |
| Lancet Psychiatry | 77.0 | 100 | 15.0; 8.6, 23.5 | 100.0; 78.1, 100.0 |
| Nature Reviews Gastroenterology and Hepatology | 73.1 | 100 | 34.0; 24.8, 44.1 | 29.4; 15.0, 47.4 |
| Lancet Public Health | 72.4 | 100 | 9.0; 4.1, 16.3 | 100.0; 66.3, 100.0 |
| Chemical Reviews | 72.1 | 100 | 7.0; 2.8, 13.8 | 57.1; 18.4, 90.1 |
| Lancet Infectious Diseases | 71.4 | 100 | 28.0; 19.4, 37.8 | 75.0; 55.1, 89.3 |
| Nature Reviews Cancer | 69.8 | 93 | 32.2; 22.9, 42.7 | 13.3; 3.7, 30.7 |
| Nature | 69.5 | 100 | 9.0; 4.1, 16.3 | 33.3; 7.4, 70.0 |
| Nature Biotechnology | 68.1 | 59 | 32.2; 20.6, 45.6 | 10.5; 1.3, 33.1 |
| Cell | 66.8 | 100 | 33.0; 23.9, 43.1 | 100.0; 89.4, 100.0 |
| Nature Reviews Clinical Oncology | 65.0 | 100 | 42.0; 32.1, 52.2 | 11.9; 3.9, 25.6 |
| Science | 63.7 | 100 | 7.0; 2.8, 13.8 | 0.0; 0.0, 40.9 |
| Chemical Society Reviews | 60.6 | 100 | 5.0; 1.6, 11.2 | 40.0; 5.2, 85.3 |
| Lancet Neurology | 59.9 | 100 | 57.0; 46.7, 66.8 | 80.7; 68.0, 89.9 |
| Nature Reviews Genetics | 59.6 | 100 | 18.0; 11.0, 26.9 | 38.8; 17.2, 64.2 |
| Lancet Oncology | 54.4 | 100 | 56.0; 45.7, 65.9 | 96.4; 87.6, 99.5 |
| Annals of Oncology | 51.7 | 100 | 77.0; 67.5, 84.8 | 98.7; 92.9, 99.9 |
| Annals of Internal Medicine | 51.6 | 100 | 26.0; 17.7, 35.7 | 19.2; 6.5, 39.3 |
| Journal of Clinical Oncology | 50.7 | 100 | 83.0; 74.1, 89.7 | 97.5; 91.5, 99.7 |
| Nature Reviews Cardiology | 49.4 | 100 | 25.0; 16.8, 34.6 | 28.0; 12.0, 49.3 |
| Nature Methods | 47.9 | 100 | 12.0; 6.3, 20.0 | 33.3; 9.9, 65.1 |
| Nature Materials | 47.6 | 91 | 5.4; 1.8, 12.3 | 0.0; 0.0, 52.1 |
| Nature Reviews Endocrinology | 47.5 | 100 | 19.0; 11.8, 28.0 | 26.3; 9.1, 51.2 |
| Physiological Reviews | 46.5 | 66 | 10.6; 4.3, 20.6 | 71.4; 29.0, 96.3 |
| Cell Research | 46.3 | 100 | 18.0; 11.0, 26.9 | 100.0; 81.4, 100.0 |
| Lancet Gastroenterology & Hepatology | 45.0 | 100 | 60.0; 49.7, 69.6 | 86.6; 75.4, 94.0 |
| Lancet Diabetes and Endocrinology | 44.8 | 100 | 54.0; 43.7, 64.0 | 96.2; 87.2, 99.5 |
| Nature Reviews Neurology | 44.7 | 100 | 26.0; 17.7, 35.7 | 19.2; 6.5, 39.3 |
| JAMA Internal Medicine | 44.4 | 100 | 30.0; 21.2, 39.9 | 43.3; 25.4, 62.5 |
| Immunity | 43.5 | 86 | 25.5; 16.7, 36.1 | 100.0; 84.5, 100.0 |
| **Total** | | **3,888** | **1,174 (30.2; 28.7, 31.6)** | **743 (63.3; 60.4, 66.0)** |

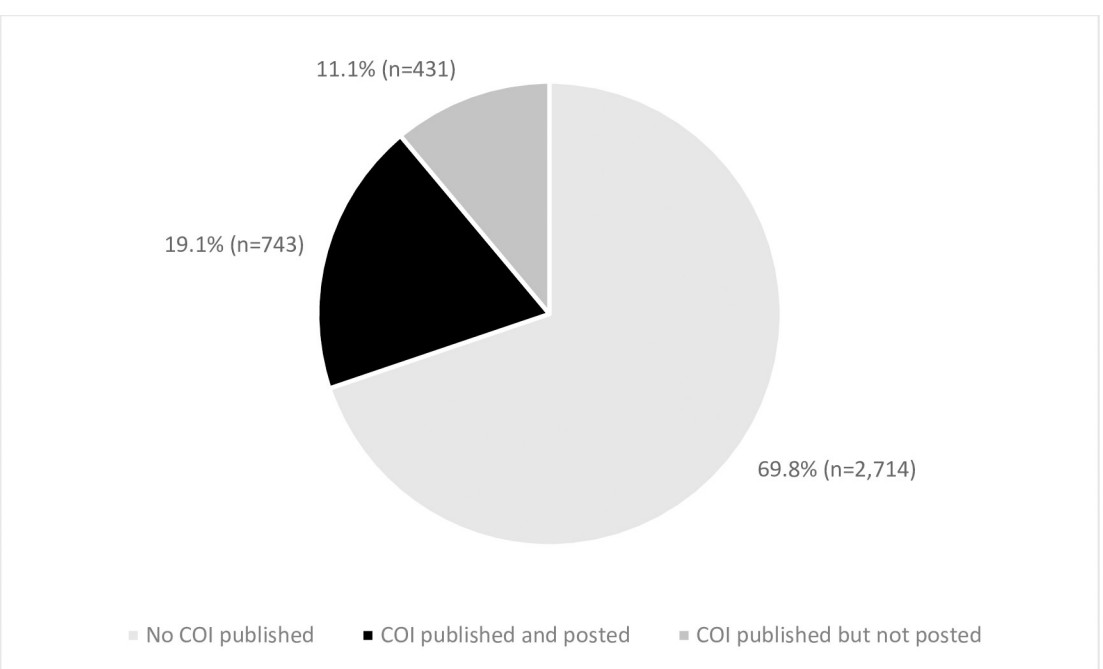

**Fig 4. Disclosure patterns of the articles in the 40 highest impact journals published from June 2021 through May 2022 and hand-searched for published COIs (n = 3,888), as of 2022.**

dependent upon both the underlying prevalence of COIs and the rate of posting. In contrast, measuring whether journals posted at least once per year eliminates the first factor as long as journals are likely to include at least one published COI in a year. Most journals publish dozens of articles each year and our analysis of the 40 highest impact journals showed that all had at least five articles with a published COI in the year studied. Thus, all journals appear to have

**Table 3. The association between article characteristics and inclusion of a posted COI statement among articles that disclosed a COI in high impact journals, June 2021 through May 2022.**

| | # (%) articles with published COIs (n = 1,174) | # (%) articles that included a posted COI statement (n = 743) | # (%) articles that did not include a posted COI statement (n = 431) | Bivariate Odds Ratio (95% CI) | Multivariate Odds Ratio (95% CI) |
|---|---|---|---|---|---|
| **Study type** | | | | | |
| **Original research** | 317 (27.0) | 266 (83.9) | 51 (16.0) | 3.74 (2.63, 5.32)*** | 5.40 (3.51, 8.32)*** |
| **Review** | 388 (33.0) | 204 (52.5) | 184 (47.4) | 0.79 (0.60, 1.04) | 0.82 (0.62, 1.08) |
| **Comment** | 469 (39.9) | 273 (58.2) | 196 (41.7) | Reference | |
| **COI location** | | | | | |
| **One location only** | 1,043 (88.8) | 653 (62.6) | 390 (37.3) | Reference | |
| **More than one location** | 131 (11.1) | 90 (68.7) | 41 (31.2) | 1.31 (0.88, 1.93) | 0.40 (0.24, 0.68)** |

The independent variables in this analysis were study type (original research/review/comment) and COI location (published COIs located under one section only (e.g., "COI statement" or "acknowledgements") (one location only)/published COIs located under a "COI statement" with additional published COIs located in the acknowledgements and/or funding section (more than one location); the dependent variable is whether or not the article included a posted COI statement; *p<0.05

**p<0.01

***p<0.001

had ample opportunity to use the field at least once in any given year and failure to do so likely indicates non-use of the field. Second, we were only able to assess the prevalence of use of the posted COI statement by articles with published COIs in the 40 highest impact journals in a single year due to the large number of articles that would have been required individual assessment had we included more journals. While this provided sufficient precision for the purposes of this study, it precludes generalization of our findings on posting rate per conflict to other journals. Third, our study reflects posted COI statements as of 2022 when we collected the data, not as of the year on which we are reporting, because posted COI statements may have been added retroactively. Fourth, while we examined the 40 highest impact journals in 2021, our findings may not be generalizable to the highest impact journals in other years. Fifth, our regression only included two covariates; additional article-level variables that were not available in this study may also be associated with the posting of COIs.

## Conclusions

Use of the PubMed COI statement has increased since it became available in 2017, but adoption is still limited, even among high impact journals. One consequence is that users of PubMed are currently unable to distinguish between a published COI that was not posted and an article with no COIs. NLM should carry out additional outreach to journals that are both not using the statement at all and using it erratically, and journals should use the statement for all COIs, regardless of where a COI is located in the paper. Until use of the statement reaches full adoption, NLM could improve the user experience through additional education of PubMed users about what is included and excluded from the COI statement and the importance of reviewing COI disclosures in the published articles.

## Acknowledgments

We would like to thank Eva Greenthal for her work on an early draft of the study protocol and Kate Peglow for her assistance with data collection. We would also like to thank staff from the National Library of Medicine for their assistance in downloading data from PubMed and for added context on the voluntary COI statement.

## Author Contributions

**Conceptualization:** Stephanie Rogus, Joseph S. Ross, Peter Lurie.

**Data curation:** Stephanie Rogus.

**Formal analysis:** Stephanie Rogus.

**Funding acquisition:** Peter Lurie.

**Methodology:** Stephanie Rogus, Joseph S. Ross, Peter Lurie.

**Supervision:** Peter Lurie.

**Writing – original draft:** Stephanie Rogus, Peter Lurie.

**Writing – review & editing:** Stephanie Rogus, Joseph S. Ross, Peter Lurie.

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
