## [Decision Letter · Decision Letter 0]

2 Jul 2024

PONE-D-24-16879Adoption of the voluntary conflict of interest statement on PubMedPLOS ONE

Dear Dr. Rogus,

Thank you for submitting your manuscript to PLOS ONE. After careful consideration, we feel that it has merit but does not fully meet PLOS ONE’s publication criteria as it currently stands. Therefore, we invite you to submit a revised version of the manuscript that addresses the points raised during the review process.

We look forward to receiving your revised manuscript.

Kind regards,

Academic Editor

PLOS ONE

 [This research was supported by a grant to CSPI from Harvey Motulsky.].  

[SR and PL have declared that no competing interests exist. JR has read the journal's policy and has the following competing interests: currently receives research support through Yale University from Johnson and Johnson to develop methods of clinical trial data sharing, from the Food and Drug Administration for the Yale-Mayo Clinic Center for Excellence in Regulatory Science and Innovation (CERSI) program (U01FD005938), from the Agency for Healthcare Research and Quality (R01HS022882), and from Arnold Ventures; formerly received research support from the Medical Device Innovation Consortium as part of the National Evaluation System for Health Technology (NEST) and from the National Heart, Lung and Blood Institute of the National Institutes of Health (NIH) (R01HS025164, R01HL144644); was an expert witness at the request of Relator's attorneys, the Greene Law Firm, in a qui tam suit alleging violations of the False Claims Act and Anti-Kickback Statute against Biogen Inc. that was settled September 2022; and is currently a Deputy Editor at JAMA, was formerly the U.S. Outreach and Research Editor at the BMJ from 2002-2003, and was formerly an Associate Editor at JAMA Internal Medicine from 2013-2019. ]. 

Additional Editor Comments:

I would suggest the authors provide a flow chart showing how study samples were selected/obtained.

In addition, kindly respond to the issues raised by the reviewers.

Reviewers' comments:

Reviewer's Responses to Questions

**Comments to the Author**

1. Is the manuscript technically sound, and do the data support the conclusions?

Reviewer #1: Yes

Reviewer #2: Yes

2. Has the statistical analysis been performed appropriately and rigorously? 

Reviewer #1: Yes

Reviewer #2: Yes

3. Have the authors made all data underlying the findings in their manuscript fully available?

Reviewer #1: No

Reviewer #2: Yes

4. Is the manuscript presented in an intelligible fashion and written in standard English?

Reviewer #1: Yes

Reviewer #2: Yes

5. Review Comments to the Author

Reviewer #1: Stephanie and colleagues have done a very good job in this area of rare interest. The have deeply searched, sieved, extracted, collated and synchronized the facts from the ocean of available information/journals. It is worth the effort put into it. The manuscript is generally well written and thus not difficult to follow through. They have also identified the key limitations which is of great importance for further./future studies It is highly commendable.

Just two peculiar./pertinent lines of thought, and thus two questions:

1. Is it possible for NLM or associated bodies or other groups of people with keen interest to alleviate this identified problem through development and use of applicable software/s?

2. Can there be any application of artificial intelligence(AI) in solving this identified problem either soon or much later in the future?

Kindest regards.

Reviewer #2: I think the statistical analysis of the manuscript was appropriately and rigorously performed and data aligned with the conclusion. Data underlying the findings were made available. The author presented the manuscript in an intelligible way and the written English appropriate.

6. PLOS authors have the option to publish the peer review history of their article (what does this mean?). If published, this will include your full peer review and any attached files.

Reviewer #1: **Yes: **Dr. Tewogbade Adeoye ADEDEJI,

Associate Professor of Chemical Pathology(Clinical Chemistry).

Department of Chemical Pathology,

College of Health Sciences,

Obafemi Awolowo University,

PMB 10, Ede road, Ile-Ife, Osun State, NIGERIA

email: tadedeji@oauife.edu.ng, philipsade@yahoo.com

Reviewer #2: **Yes: **IBRAHIM Rabiu

---

## [Author Response · Author response to Decision Letter 0]

12 Jul 2024

Our response to the editor and reviewers is enclosed in the appropriate attached file.

---

## [Decision Letter · Decision Letter 1]

31 Jul 2024

Adoption of the voluntary conflict of interest statement on PubMed

PONE-D-24-16879R1

Dear Dr. Lurie,

We’re pleased to inform you that your manuscript has been judged scientifically suitable for publication and will be formally accepted for publication once it meets all outstanding technical requirements.

Kind regards,

Tope Michael Ipinnimo, MBBS, MPH, FWACP, FMCPH

Academic Editor

PLOS ONE

Additional Editor Comments (optional):

Reviewers' comments:

Reviewer's Responses to Questions

**Comments to the Author**

1. If the authors have adequately addressed your comments raised in a previous round of review and you feel that this manuscript is now acceptable for publication, you may indicate that here to bypass the “Comments to the Author” section, enter your conflict of interest statement in the “Confidential to Editor” section, and submit your "Accept" recommendation.

Reviewer #1: All comments have been addressed

2. Is the manuscript technically sound, and do the data support the conclusions?

Reviewer #1: Yes

3. Has the statistical analysis been performed appropriately and rigorously? 

Reviewer #1: Yes

4. Have the authors made all data underlying the findings in their manuscript fully available?

Reviewer #1: Yes

5. Is the manuscript presented in an intelligible fashion and written in standard English?

Reviewer #1: Yes

6. Review Comments to the Author

Reviewer #1: No further comments to the author. No further comments to the authors; No further comments to the authors

7. PLOS authors have the option to publish the peer review history of their article (what does this mean?). If published, this will include your full peer review and any attached files.

Reviewer #1: **Yes: **Adedeji Tewogbade Adeoye

---

## [Editor Report · Acceptance letter]

21 Oct 2024

PONE-D-24-16879R1 

PLOS ONE

Dear Dr. Lurie, 

I'm pleased to inform you that your manuscript has been deemed suitable for publication in PLOS ONE. Congratulations! Your manuscript is now being handed over to our production team.

Kind regards, 

on behalf of

Dr. Tope Michael Ipinnimo 

Academic Editor

PLOS ONE